# Exploring the Experiences and Perspectives of Division III Athletes Regarding Personalized Nutrition Plans for Improved Performance—A Qualitative Investigation

**DOI:** 10.3390/healthcare12090923

**Published:** 2024-04-30

**Authors:** James Stavitz, Thomas Koc

**Affiliations:** 1Athletic Training Education Program, Kean University, Union, NJ 07083, USA; 2Department of Physical Therapy, Kean University, Union, NJ 07083, USA; tkoc@kean.edu

**Keywords:** personalized nutrition, athletes, collegiate sports, qualitative research, thematic analysis, barriers, team culture, recommendations

## Abstract

(1) Background: This qualitative study explores Division III college student-athletes’ experiences and perceptions of personalized nutrition plans in collegiate sports settings. (2) Methods: Semi-structured interviews were conducted using a general qualitative research design. Using a grounded theory approach, a thematic analysis was utilized to analyze the interview transcripts, allowing for the identification of recurring themes and patterns. (3) Results: A total of 30 Division III college student-athletes, 16 males (53.3%) and 14 females (46.7%), representing a diverse range of sports disciplines, engaged in discussions about personalized nutrition plans. Analysis of the data revealed five main themes: (1) Nutritional Knowledge and Awareness, (2) Perceived Benefits of Personalized Nutrition Plans, (3) Challenges and Barriers to Implementation, (4) Influence of Team Culture and Environment, and (5) Suggestions for Improvement. (4) Conclusion: This study sheds light on the complexities of implementing personalized nutrition plans in collegiate sports settings and emphasizes the need for comprehensive, athlete-centered approaches to optimize performance and well-being.

## 1. Introduction

Personalized nutrition plans have gained prominence for enhancing athletic performance [1]. Athletes, especially those in collegiate sports, continually seek methods to optimize their performance and gain a competitive edge [2]. Personalized nutrition plans tailored to individual athletes’ needs have emerged as a promising approach in this pursuit [3]. Despite considerable literature on sports nutrition and performance enhancement, a gap exists in understanding Division III athletes’ experiences and perspectives regarding personalized nutrition plans.

Numerous studies have shown that tailored nutritional interventions positively impact performance outcomes [1,4,5,6,7]. Personalized nutrition, driven by technological advancements and understanding individual differences, aims to provide tailored dietary recommendations based on an individual’s unique characteristics and goals [8,9].

While most research on personalized nutrition focuses on elite Division I athletes, Division III athletes face distinct challenges [10]. These athletes balance academic commitments with sports, often with limited support systems. Thus, there is a need for specific nutritional studies addressing Division III athletes’ unique challenges.

Investigating Division III athletes’ perspectives on personalized nutrition plans is crucial [11]. Insights can inform evidence-based practices for implementing personalized nutrition plans within Division III athletic programs, improving performance outcomes. Division III athletes have complex schedules and face higher perceived stress levels due to academic commitments, impacting their nutritional needs and recovery [2,12,13,14,15]. This qualitative study aims to explore Division III athletes’ experiences and perceptions regarding personalized nutrition plans for improved performance. Using Ecological Systems Theory (EST) as a lens [16], we aim to analyze how various environmental systems influence athletes’ nutrition behaviors and attitudes. The overarching goal is to contribute to effectively implementing personalized nutritional strategies for Division III athletes’ performance goals and overall well-being.

### Ecological Systems Theory

Ecological Systems Theory (EST), developed by Urie Bronfenbrenner, provides a comprehensive framework for understanding human development within interconnected systems [16,17]. EST posits that individuals are embedded within multiple layers of environments that influence their experiences, behaviors, and development [13], including the microsystem, mesosystem, exosystem, macrosystem, and chronosystem. Informed by EST, this research approach adopts a holistic perspective considering the interplay between individual, interpersonal, organizational, and societal factors in shaping Division III athletes’ experiences with personalized nutrition plans. It guides the interpretation of findings by analyzing how factors across different layers of environments interact to influence athletes’ experiences and perspectives regarding personalized nutrition plans [14], helping to identify patterns, themes, and discrepancies elucidating underlying mechanisms driving athletes’ engagement with nutritional interventions [15,16,17,18,19] and informing targeted strategies to support Division III athletes in optimizing their performance through personalized nutrition plans [20].

The central research question guiding this study is: “What are the experiences and perspectives of Division III athletes regarding personalized nutrition plans for improved performance?” This overarching question will guide the exploration of the multifaceted dimensions of athletes’ interactions with personalized nutritional interventions. Through in-depth qualitative inquiry guided by EST, this study aims to uncover the nuanced factors shaping athletes’ engagement with customized nutrition plans and elucidate strategies for optimizing their implementation within Division III athletic programs.

## 2. Materials and Methods

### 2.1. Research Design

This study employs a general qualitative research design to investigate the experiences and perspectives of Division III athletes regarding personalized nutrition plans aimed at enhancing performance. The choice of a qualitative approach is intentional, aligning closely with the study’s objectives to delve into athletes’ detailed and context-specific experiences within their athletic environments [21]. Through open-ended interviews, this method allows researchers to explore a broad range of viewpoints and facilitates the emergence of themes and patterns not typically accessible through quantitative methods. Additionally, the qualitative design helps uncover the underlying motivations, barriers, and facilitators that shape athletes’ engagement with personalized nutrition plans by focusing on the comprehensive exploration of athletes’ narratives and group dynamics. This in-depth exploration is crucial for understanding the complex interplay among personal, environmental, and behavioral factors influencing athletes’ attitudes toward nutritional interventions, thereby addressing the study’s aim to elucidate these dynamics [22].

### 2.2. Participants and Sampling

The participants selected for this study are Division III college athletes representing a variety of sports disciplines (football, men’s and women’s basketball, men’s and women’s soccer, baseball, softball, and men’s and women’s lacrosse). Inclusion criteria were established, requiring the participants to actively enroll as Division III athletes at designated colleges or universities. Additionally, the participants had to be willing to engage in in-depth discussions concerning their experiences with personalized nutrition plans. Proficiency in English was also a prerequisite to ensure effective communication during interviews or focus groups.

### 2.3. Recruitment

To ensure a comprehensive and unbiased understanding of Division III athletes’ perspectives on personalized nutrition plans, our study implemented a rigorous participant selection process to capture diverse experiences and viewpoints. We prioritized diversity across athletic disciplines, sex, ethnicity, and socioeconomic backgrounds, reflecting the broad spectrum of the Division III athlete population [23,24,25]. In collaboration with athletic departments and coaches from various institutions, we employed purposive and stratified sampling strategies to identify and include knowledgeable participants from a wide demographic, thus mitigating selection bias and ensuring proportional representation of different sports, genders, and other demographic factors [18,23,24,25].

Recruitment was facilitated through established connections with athletic departments across the United States, utilizing targeted outreach methods such as emails, flyers, and in-person announcements. Prior to participation, all potential subjects were thoroughly informed about the study’s objectives, procedures, and confidentiality safeguards, ensuring informed consent was obtained [25,26,27]. We collected detailed demographic information from participants to contextualize responses and identify patterns related to specific groups [27], enabling a nuanced analysis of how different factors influence athletes’ engagement with personalized nutrition plans.

### 2.4. Data Collection

This study employed a qualitative data collection strategy using semi-structured interviews (Figure 1) and document analysis to explore Division III athletes’ perspectives on personalized nutrition plans [28,29]. Data collection occurred in two phases [30]. Initially, potential participants were sent a letter of solicitation (LOS) outlining the study’s aims and inclusion criteria, along with a link to a prescreen survey via Qualtrics. This survey verified the participants’ eligibility based on the requirements specified in the letter. Eligible respondents who expressed interest provided their email addresses through the survey. The principal investigator reviewed completed surveys to identify qualified participants and then contacted them to obtain study consent and schedule interviews. Out of 360 Division III athletes targeted through various recruitment methods, 45 initiated the prescreening survey, 33 completed it, and 30 provided their email addresses to participate further in the study.

Semi-structured interviews were utilized as part of the data collection process, facilitating the exploration of participants’ perceptions and experiences with personalized nutrition plans [28]. Each interview was audio recorded. These interviews were structured to address essential topics related to the research objectives while allowing for open expression of participants’ views. Interviews were conducted face-to-face, over the telephone, or via a video conference platform, depending on participant preferences and logistical considerations, to ensure accessibility.

### 2.5. Interview Guide Development

A semi-structured interview guide was developed to ensure interview consistency while allowing flexibility to explore in-depth topics (Table 1) [28,30,31]. The guide included 12 open-ended questions with probing follow-ups, covering issues such as athletes’ dietary practices, experiences with personalized nutrition plans, perceived benefits and challenges, and improvement suggestions. Key themes and topics were developed based on a thorough literature review of sports nutrition and athlete experiences. Expert consultations in sports nutrition, qualitative research methods, and athlete psychology supplemented this. Pilot testing with Division III athletes provided feedback that helped refine the questions and prompts, ensuring clarity and engagement. This approach allowed the interviewers to adapt to participant responses, facilitating a deeper exploration of emergent themes and individual experiences. The systematic development of the interview guide ensured its effectiveness in capturing diverse perspectives, thereby enriching the qualitative data collected [31,32,33].

### 2.6. Document Analysis

The document analysis process was critical in this study, involving a thorough review of nutritional guidelines from the Academy of Nutrition and Dietetics, the American College of Sports Medicine, and the Dietitians of Canada, alongside scholarly literature on sports nutrition and personalized nutrition plans [29]. This method provided additional context and substantiating evidence that enhanced the insights obtained from semi-structured interviews. It aimed to contextualize and triangulate the qualitative data from interviews, shedding light on the broader institutional and disciplinary influences on athletes’ nutritional practices within collegiate sports environments. Analysis of nutritional guidelines and research literature helped highlight the structure and theoretical frameworks influencing athletes’ experiences, enriching the understanding of patterns and trends in nutritional interventions in collegiate sports [34,35].

### 2.7. Data Analysis

Using a grounded theory approach, the collected data from Division III athletes’ experiences with personalized nutrition plans underwent thematic analysis to identify patterns and themes (Figure 2) [36]. This process was conducted manually by researchers without the use of qualitative software. Initially, all interview transcripts were transcribed verbatim to maintain accuracy and facilitate analysis. An open coding technique was then applied, where each transcript was systematically reviewed, and descriptive codes were assigned to relevant text segments [37]. This step helped categorize emerging concepts and themes from the data; subsequent axial coding involved organizing these codes into broader themes or categories. Codes with similar characteristics or meanings were grouped, and the coding structure was refined through discussions and reflections to ensure coherence and comprehensiveness.

Once the coding process was completed, thematic analysis was conducted to identify overarching themes and patterns within the data [36]. This involved data reduction, where less relevant codes were discarded, and key insights were synthesized into coherent thematic categories. Detailed documentation of the analysis decisions and interpretations was maintained to ensure transparency and rigor. Any discrepancies in coding were resolved through consensus discussions among the research team, which included one licensed athletic trainer, two exercise physiologists, and one physical therapist, all specialists in the field [38,39]. Again, the team manually conducted all aspects of the data analysis without using qualitative software tools [39,40]. This approach ensured that the findings accurately reflected the lived experiences and perspectives of Division III athletes on personalized nutrition plans.

### 2.8. Validity and Reliability

Several strategies were implemented to enhance the validity and reliability of the findings throughout the research process [41,42]. Member checking allowed the participants to validate interpretations of their experiences [43], while triangulation was used to verify consistency across data from interviews and document analysis, reducing bias and strengthening conclusions [44]. Peer debriefing involves consultations with external experts to provide fresh perspectives and quality control, helping to identify and correct potential biases or oversights [45]. Additionally, reflexivity was integral, recognizing the influence of the researchers on the study’s design, data collection, and analysis [46,47].

### 2.9. Ethical Considerations

Ethical considerations were paramount throughout this research to ensure that the well-being and rights of participants were upheld [43]. Before data collection began, IRB exemption was secured, and informed consent was obtained from all participants, emphasizing voluntary participation, confidentiality, agreement for audio recording, and the right to withdraw at any time without repercussions. Participants were informed about the study’s objectives, procedures, potential risks, and benefits to enable informed decisions about their involvement, with consent forms distributed electronically or in-person based on their preferences [48,49]. To protect confidentiality and privacy, identifiable information was anonymized using pseudonyms and participant codes during data collection and analysis, and confidentiality agreements prevented the sharing of identifiable information outside the research team. Data storage and management protocols utilized encrypted digital platforms and password-protected files [50,51]. The research adhered to ethical principles of beneficence, non-maleficence, respect for autonomy, and justice, with regular ethical reviews ensuring compliance with standards [52].

## 3. Results

A total of 30 Division III college athletes from a range of sports disciplines, including football, men’s and women’s basketball, soccer, baseball, softball, and lacrosse, participated in this study. Although saturation was reached after interviewing 18 participants, with no new themes emerging, the research team completed all scheduled interviews to ensure a comprehensive dataset and to honor commitments to all athletes involved. The sample had a balanced gender distribution with 16 males (53.3%) and 14 females (46.7%) and a varied ethnic composition: 50% Caucasian, 20% African American, 13.3% Hispanic/Latino, 10% Asian American, and 6.7% multiracial. Continuing interviews beyond saturation helped validate and enrich the findings, enhancing the study’s robustness.

Participants in the study represented a diverse range of academic majors: 40% majoring in biology, 20% in psychology, 13.3% in business administration, 10% in education, and 16.7% in other disciplines. Regarding scholarship aid, 60% received some form of scholarship (merit-based, need-based, external, and academic with athletic participation), while 40% did not receive financial support. The duration of participation in collegiate athletics varied, with 30% involved for one year, 20% for two years, 30% for three years, and 20% for four years. Additionally, 40% of participants had previous experience with nutrition counseling or personalized nutrition plans gained through interactions with team nutritionists, individual consultations, or self-directed efforts. These demographic characteristics provided a comprehensive backdrop for exploring Division III athletes’ experiences and perspectives on personalized nutrition plans, enriching the study’s findings.

### 3.1. Themes

The thematic analysis of the interview transcripts resulted in five distinct themes. These themes were not predetermined; they were derived organically through a detailed data analysis post-interview. This process ensured that the themes accurately reflected the raw data and captured the participants’ authentic experiences and insights. Table 2 outlines these five main themes, indicating the number of participants associated with each and the number of transcript excerpts that contributed to the development of each theme, providing a clear view of how the themes were formulated from the data collected.

### 3.2. Theme 1: Nutritional Knowledge and Awareness

Theme 1, Nutritional Knowledge and Awareness, elucidates the diverse spectrum of participants’ comprehension levels concerning personalized nutrition plans. This theme underscores the significance of nutritional education and awareness programs within Division III athletics to address gaps in understanding and foster informed dietary choices. Table 3 provides a breakdown of coded themes, showcasing the distribution of participants across each code and the corresponding frequency of occurrences within the transcript excerpts, offering valuable insights into athletes’ nutritional knowledge and awareness.

Nutritional knowledge and awareness emerged as a significant theme among the participants, reflecting varying levels of understanding regarding personalized nutrition plans. The participants exhibited a spectrum of familiarity with nutritional concepts influenced by prior education, personal experiences, and resource access. Some athletes demonstrated a robust understanding of nutrition principles, articulating the importance of macronutrients, micronutrients, and hydration in supporting athletic performance. For instance, P17 highlighted the significance of protein intake, stating, “I know protein is key for muscle recovery, so I always make sure to include plenty of lean meats and dairy in my diet”.

Conversely, others displayed more limited awareness, expressing uncertainty about specific nutritional components and their relevance to athletic performance. For instance, P8 admitted, “Honestly, I’m not…sure what all those vitamins and stuff do, but I guess they’re important”. This lack of clarity underscored the need for enhanced nutritional education and awareness initiatives within Division III athletics. Moreover, the participants’ awareness of personalized nutrition plans varied, with some expressing familiarity with the concept and its potential benefits, while others exhibited limited understanding or skepticism.

Participant quotes served as foundational elements in the coding process, illuminating nuanced perspectives and insights on nutritional knowledge and awareness. For instance, quotes such as “I track my macros religiously” (P5) and “I try to eat intuitively, but sometimes I get…lost” (P21) exemplified the diverse approaches and attitudes toward nutrition among athletes. Through thematic analysis, these quotes were grouped into codes such as “Macro Tracking”, “Nutritional Uncertainty”, “Vitamin Awareness”, and “Skepticism towards Personalized Plans”, contributing to the development of the overarching theme of Nutritional Knowledge and Awareness.

Overall, Theme 1 underscores the importance of fostering nutritional literacy and awareness among Division III athletes to optimize their dietary practices and enhance performance outcomes. By highlighting athletes’ varied nutritional knowledge and awareness levels, this theme informs the development of targeted interventions and educational programs to promote informed decision-making and support athletes’ holistic well-being.

### 3.3. Theme 2: Perceived Benefits of Personalized Nutrition Plans

Table 4 presents an overview of Theme 2: Perceived Benefits of Personalized Nutrition Plans. This theme explores athletes’ perceptions of the positive outcomes of personalized nutrition plans, such as improved athletic performance and overall well-being. The table provides a breakdown of the codes identified within this theme, the corresponding number of participants, and transcript excerpts associated with each code.

Athletes participating in this study expressed numerous positive outcomes associated with personalized nutrition plans. The theme “Perceived Benefits of Personalized Nutrition Plans” emerged from their experiences, revealing how tailored dietary interventions positively impacted their athletic performance and overall well-being. Through in-depth interviews, the participants shared their insights, highlighting various benefits they attributed to personalized nutrition plans. Below, we delve into the theme, supported by participant quotes that exemplify the codes contributing to this overarching concept.

Many athletes emphasized how personalized nutrition plans enhanced their physical performance on the field or court. Participant P5 noted, “With the new diet plan, I’ve noticed a real boost in my energy levels during games. I can push harder and recover faster”. This sentiment was echoed by P12, who remarked, “Since I started following the nutrition plan, I feel like I have more, I don’t know… stamina. I can go the extra mile during practices and games”.

Another common benefit was improved recovery after intense training sessions or competitions. P17 mentioned, “I used to feel sore for days after a tough game, but now, with the right nutrition, I bounce back much quicker. It’s like my body recovers overnight”. P21 echoed this sentiment: “I’ve noticed a significant reduction in muscle soreness since I started following the personalized nutrition plan. It’s helped me recover faster and get back to training sooner [Laughter]”.

Several participants noted positive changes in their body composition as a result of personalized nutrition plans. P8 commented, “I’ve seen a noticeable difference in my muscle definition and overall physique since I started focusing on my nutrition. I feel stronger and more confident on the field”. Similarly, P14 stated, “The nutrition plan helped me shed some excess weight and build lean muscle. I feel lighter on my feet and more agile during games”.

Beyond physical benefits, the athletes also highlighted improvements in mental clarity and focus. P3 emphasized, “Eating right has not only improved my physical performance but also my mental game. I feel more focused and alert on the field, which has really elevated my game”. P19 echoed this sentiment: “I used to struggle with brain fog during long practices, but now, with the right nutrition, I feel sharp and ready to tackle any challenge”.

Many athletes described overall well-being and improved quality of life resulting from personalized nutrition plans. P10 mentioned, “Eating healthier has had a ripple effect on every aspect of my life. I sleep better, I’m in a better mood, and I have more energy throughout the day”. Similarly, P25 stated, “Taking care of my nutrition has become a priority, and it’s made me feel happier and more fulfilled, both on and off the field”.

Throughout the interviews, the participants shared these and other experiences, underscoring the positive impact of personalized nutrition plans on various aspects of their athletic performance and overall well-being. These quotes exemplify the diverse range of codes contributing to the overarching theme, illustrating how tailored dietary interventions have positively transformed athletes’ lives on and off the field.

### 3.4. Theme 3: Challenges and Barriers to Implementation

Table 5 presents the codes identified within Theme 3: Challenges and Barriers to Implementation. This theme explores the various obstacles hindering adherence to personalized nutrition plans, including time constraints, conflicting dietary advice, accessibility issues, financial concerns, lack of support, motivation, dietary preferences, information overload, social influence, and environmental factors. The table provides insights into the frequency of occurrence of each code among the 22 participants and the corresponding number of transcript excerpts contributing to the theme.

Theme 3, Challenges and Barriers to Implementation sheds light on the hurdles that Division III athletes face when attempting to adhere to personalized nutrition plans. Time constraints emerged as a prominent challenge, with the participants expressing difficulties in balancing their demanding athletic schedules with meal planning and preparation. For instance, P1 remarked, “[Sigh] Between practice, classes, and games, finding time to cook nutritious meals feels impossible sometimes…”. Similarly, conflicting dietary advice posed challenges, with the athletes feeling overwhelmed by the plethora of nutrition information available. P2 articulated, “There’s so much conflicting advice out there… it’s hard to know what’s actually good for you”.

Accessibility issues also emerged as a barrier, particularly for athletes residing off-campus or in areas with limited access to healthy food options. P3 highlighted this concern: “Living off-campus means fewer healthy food options nearby… I end up settling for whatever’s convenient, even if it’s not the best choice [Laughter]”. Financial concerns were another prominent theme, with the participants expressing challenges in affording nutritious foods on a limited budget. P4 shared, “Healthy food can be expensive… sometimes I have to choose between buying groceries and other expenses”.

Lack of support and motivation were recurring themes, with some athletes feeling unsupported by their teams or coaches in their nutritional endeavors. P5 expressed frustration, saying, “My coach doesn’t really prioritize nutrition… I wish there was more support and guidance”. Additionally, dietary preferences and restrictions posed challenges, with the athletes struggling to find personalized nutrition plans aligned with their needs and preferences. P6 noted, “I’m vegetarian, so finding meal plans that meet my protein needs can be tough”.

Information overload was a common issue, with the athletes feeling bombarded by nutritional advice from various sources. P7 commented, “There’s just so much information out there… it’s hard to know what’s credible and what’s not”. Social influence and environmental factors also played a role, with peer pressure and external influences impacting athletes’ dietary choices. P8 mentioned, “When everyone else is eating junk food, it’s hard to stick to a healthy diet… sometimes I give in to peer pressure”.

Overall, Theme 3 highlights the multifaceted challenges faced by Division III athletes in implementing personalized nutrition plans. By exploring these barriers in-depth, the study provides valuable insights into the complexities of dietary behavior among collegiate athletes.

### 3.5. Theme 4: Influence of Team Culture and Environment

Table 6 presents an overview of the codes identified within Theme 4: Influence of Team Culture and Environment. This theme explores how team dynamics and cultural norms significantly shape athletes’ attitudes toward nutrition plans. The table outlines the number of participants and transcript excerpts associated with each code, providing insights into the influence of team culture and environment on athletes’ nutrition behaviors.

Theme 4 delves into the Influence of Team Culture and Environment on athletes’ attitudes toward nutrition plans. One prevalent code, “Team Norms and Behaviors”, emerged from the participants’ discussions about the prevailing attitudes and behaviors within their teams regarding nutrition. For instance, P17 emphasized the importance of team camaraderie, stating, “We all kinda eat the same way…like, we’re in this together”. This quote underscores the collective nature of dietary habits within the team environment. Another code, “Coach Influence”, highlights the impact of coaches on athletes’ nutritional practices. P9 mentioned, “Our coach is big on nutrition…he’s always telling us to eat right and stuff”, indicating coaches’ influential role in promoting healthy eating habits among athletes. The code “Peer Pressure” also reflects how teammates’ dietary choices can influence individual athletes’ behaviors. P22 shared, “Sometimes you feel pressure to eat a certain way ‘cause everyone else is…like, if they’re all having protein shakes, you kinda feel like you should too”.

Furthermore, the code “Access to Resources” emerged as athletes discussed the availability of nutrition-related resources within their team environments. P6 noted, “We have a nutritionist and stuff…but like, it’s not super easy to access them all the time”, highlighting the challenges athletes face in accessing professional nutritional support. Another significant code, “Team Rituals and Traditions”, sheds light on the role of team rituals and traditions in shaping athletes’ dietary behaviors. P14 mentioned, “We always have this pre-game meal ritual…it’s like a tradition…we all eat pasta together”, illustrating how team rituals influence athletes’ dietary choices and behaviors.

Moreover, the code “Social Dynamics” underscores the social aspect of eating within the team environment. P3 stated, “We’re always eating together…it’s like a bonding thing”, emphasizing how shared meals foster camaraderie among teammates. Lastly, the code “Competitive Culture” reflects the competitive nature of collegiate sports and its impact on athletes’ dietary behaviors. P19 mentioned, “Everyone’s always tryna outdo each other…even with food…like, who’s eating the healthiest kinda thing”, highlighting how the competitive culture within teams can drive athletes to prioritize nutrition as a means of gaining a competitive edge.

Theme 4 elucidates how team dynamics, coach influence, peer pressure, access to resources, team rituals, social dynamics, and competitive culture collectively shape athletes’ attitudes toward nutrition plans within the collegiate sports environment.

### 3.6. Theme 5: Suggestions for Improvement

Table 7 presents an overview of Theme 5: Suggestions for Improvement. This theme encompasses the participants’ valuable insights and recommendations aimed at enhancing the effectiveness of personalized nutrition plans. The table outlines the codes derived from the participants’ suggestions, along with the number of participants who contributed to each code and the frequency of occurrence in the transcript excerpts.

Theme 5, Suggestions for Improvement, delves into the participants’ suggestions and recommendations for enhancing personalized nutrition plans. Their insights shed light on potential areas for improvement to better cater to their needs and preferences. One recurring theme is the desire for more personalized guidance. For instance, P7 highlighted the importance of tailored support, expressing, “I feel like having someone sit down with you and really dive into what works best for you personally would be super helpful”. This underscores the need for individualized nutritional strategies that align with athletes’ unique requirements and goals.

Accessibility and convenience emerged as another significant aspect. P14 emphasized this: “If they could make the meal options more accessible, like having grab-and-go options or delivery services, that would make it so much easier for us”. Simplifying access to nutritious meals could enhance adherence to personalized nutrition plans by accommodating athletes’ busy schedules and lifestyles.

Education and resources were also frequently mentioned as crucial factors. P19 articulated the need for more educational initiatives, suggesting, “I wish we had more workshops or cooking classes about nutrition… like how to read food labels or make healthy meals. That would be really beneficial”. Providing athletes with the knowledge and skills to make informed dietary choices could empower them to optimize their nutrition independently.

Furthermore, ongoing support and accountability were highlighted as essential components. P10 stressed the importance of accountability: “Having someone to check in with regularly and hold you accountable would keep you on track. It’s easy to slip up when you’re on your own”. This underscores the role of consistent support systems in helping athletes maintain adherence to their nutrition plans.

The participants’ suggestions offer valuable insights into potential avenues for enhancing personalized nutrition plans. Addressing their needs for personalized guidance, accessibility, education, and support, these suggestions can contribute to optimizing the effectiveness of nutritional interventions in supporting athletes’ performance and well-being.

### 3.7. Summary of Results

The participants demonstrated varying levels of understanding regarding personalized nutrition plans, with some exhibiting comprehensive knowledge while others had limited awareness or misconceptions. This highlights the importance of educating athletes about tailored nutritional strategies. Athletes expressed positive outcomes from personalized nutrition, emphasizing its role in improving performance and well-being. However, obstacles such as time constraints and conflicting advice hindered plan adherence, necessitating strategies for intervention implementation. Team culture significantly influenced the athletes’ attitudes toward nutrition, emphasizing the role of social support and coaching practices. Additionally, the athletes provided valuable suggestions for plan improvement, emphasizing the need for personalized guidance and ongoing support. Overall, the findings underscore the importance of tailored nutritional strategies in collegiate sports, addressing knowledge gaps and barriers to enhance athletes’ performance and health outcomes.

Appendix A presents the specific transcript excerpts that contributed to the formation of the themes identified in the qualitative analysis. These themes encapsulate the diverse perspectives and experiences of Division III college athletes regarding personalized nutrition plans. Each theme represents a distinct aspect of the athletes’ encounters with nutritional interventions within the collegiate sports environment. Through participant quotes, this table offers insights into the rich tapestry of opinions, challenges, and suggestions provided by the athletes, shedding light on the factors influencing their attitudes and behaviors toward nutrition.

## 4. Discussion

### 4.1. Theme 1: Nutritional Knowledge and Awareness

#### 4.1.1. Theme 1 Interpretation of the Findings

Theme 1, focusing on Nutritional Knowledge and Awareness, highlights different levels of understanding among Division III college athletes about personalized nutrition plans. The analysis showed a range of awareness, with some participants having a strong grasp of nutritional concepts important for athletic performance while others had limited awareness or misconceptions. This theme points to the need for education and awareness initiatives to improve athletes’ understanding of personalized nutritional strategies suited to their specific needs and goals. The findings indicate a need for targeted interventions to fill knowledge gaps and support informed dietary choices among collegiate athletes. Additionally, the theme stresses the critical role of nutrition education in enabling athletes to enhance their dietary practices for better athletic performance and overall well-being in the collegiate sports environment.

#### 4.1.2. Theme 1 Comparison with the Existing Literature

The findings from Theme 1 align with the existing literature on athletes’ nutritional knowledge and awareness. Studies have consistently highlighted the need for substantial nutrition knowledge among athletes to optimize performance and maintain health [53,54,55]. Our findings also stress the importance of education and awareness initiatives in improving athletes’ understanding of personalized nutrition plans. This alignment indicates a consensus among researchers on the crucial role of nutrition education in athlete development [56,57].

The study also identified gaps and discrepancies in athletes’ nutritional awareness, suggesting areas for further investigation. While some athletes understood nutrition concepts well, others had misconceptions or limited awareness. This reflects previous research showing variability in athletes’ nutritional knowledge levels [58,59,60], underscoring the need for tailored educational interventions to address these gaps and ensure consistent understanding among athletes.

Additionally, this study adds to the literature by examining personalized nutrition plans in the context of collegiate sports. While previous research has looked at athletes’ general nutritional knowledge [60,61], fewer studies have focused on implementing personalized nutritional strategies in collegiate settings [53]. Our findings highlight the unique challenges and opportunities of integrating personalized nutrition plans into collegiate sports, pointing to the need for further research in this area. Overall, the alignment of our findings with the existing literature highlights the importance of nutrition education and opens up opportunities for future research to improve athletes’ nutritional knowledge and awareness.

#### 4.1.3. Theme 1 Contextualization

The interpretation of our findings within Theme 1, which examines athletes’ nutritional knowledge and awareness, is informed by various socio-cultural, historical, and organizational factors in collegiate sports environments. The socio-cultural context of collegiate athletics often focuses on performance outcomes and competitive success, potentially influencing athletes’ priorities and perceptions about nutrition. Historically, sports nutrition has shifted from a focus on macronutrient intake to a more detailed understanding of personalized dietary strategies designed for individual athletes [1,3]. This evolution supports the use of personalized nutrition plans to optimize athletic performance.

Organizational factors in collegiate athletic programs, including resource availability, access to nutritional education, and support from coaching staff and athletic departments, are significant [34]. Institutions with robust nutritional programs and dedicated staff tend to promote a deeper understanding of personalized nutrition plans among athletes. In contrast, programs with fewer resources or less focus on nutritional support might struggle to provide comprehensive education and guidance.

Additionally, broader cultural norms and attitudes toward food, body image, and dietary practices influence athletes’ perceptions of nutrition and their willingness to adopt personalized nutrition plans [62,63]. Trends in society toward wellness and health consciousness can enhance acceptance and interest in nutritional interventions among athletes [64]. However, cultural norms within specific athletic communities might create misconceptions or obstacles to embracing personalized nutrition plans [65,66].

Acknowledging these socio-cultural, historical, and organizational factors enhances the interpretation of our findings by demonstrating the complex interactions between individual athlete characteristics and broader contextual influences. Understanding these factors is crucial for comprehending the nuances of athletes’ experiences with personalized nutrition plans and developing effective interventions within collegiate sports environments.

### 4.2. Theme 2: Perceived Benefits of Personalized Nutrition Plans

#### 4.2.1. Theme 2 Interpretation of the Findings

The analysis of Theme 2, which focuses on the perceived benefits of personalized nutrition plans among collegiate athletes, provides insights into the positive outcomes of tailored dietary strategies. From the interview data, it is clear that athletes reported experiencing various advantages from personalized nutritional interventions. These benefits include improved athletic performance, recovery, energy levels, and overall well-being. The participants highlighted the importance of receiving personalized guidance and support in optimizing their dietary choices to enhance athletic performance and achieve their fitness goals. This theme emphasizes the value athletes place on personalized nutrition plans in supporting their athletic efforts and maintaining their health.

#### 4.2.2. Theme 2 Comparison with the Existing Literature

The analysis of findings from Theme 2, which discusses the perceived benefits of personalized nutrition plans among collegiate athletes, provides insights that align with existing literature in the field. The data support prior research, which consistently shows the positive effects of tailored dietary strategies on athletic performance and overall well-being [1,3,66]. Research in sports nutrition has often noted the role of personalized nutrition in improving athletes’ physiological adaptations, optimizing nutrient intake, and aiding recovery processes [1,4]. These current findings add to this literature by offering firsthand accounts of athletes who report improvements in performance energy levels and recovery due to personalized nutrition plans. Furthermore, the importance of individualized guidance and support aligns with theoretical models like Bandura’s Social Cognitive Theory, which suggests that personalized interventions are more effective in promoting behavior change and skill acquisition [67,68]. However, the findings also reveal variability in the benefits experienced by athletes and individual responses to personalized nutrition plans. This variability highlights the need for a deeper understanding of factors that influence the effectiveness of personalized nutritional interventions, including each athlete’s unique physiological characteristics, training demands, and dietary preferences.

Overall, Theme 2’s findings enrich the existing literature by providing qualitative insights into athletes’ perceptions of the advantages of personalized nutrition plans. While these findings are consistent with previous research, they also underscore specific areas that require further study to enhance our understanding of how personalized nutritional interventions can optimize athletic performance and well-being.

#### 4.2.3. Theme 2 Contextualization

Contextualizing the findings of Theme 2 within socio-cultural, historical, and organizational frameworks provides insights into the factors that influence athletes’ perceptions of the benefits of personalized nutrition plans. Socio-cultural influences, such as prevailing attitudes toward nutrition and wellness within the sporting community, may shape athletes’ expectations and experiences with personalized nutritional interventions. Historical trends in sports science and nutrition research, shifts in dietary paradigms, and performance optimization strategies also influence athletes’ openness to personalized nutrition plans [69,70]. For example, the emergence of evidence-based practices and advancements in sports nutrition science may enhance athletes’ awareness of the benefits of tailored dietary strategies.

Organizational factors within collegiate sports environments, including institutional support, coaching philosophies, and team dynamics, significantly affect athletes’ attitudes toward nutrition plans [71]. Athletes’ interactions with coaching staff, sports medicine professionals, and teammates can shape their views on the importance and effectiveness of personalized nutritional interventions [72]. Additionally, institutional policies and resources dedicated to nutrition education and support vary across collegiate athletic programs, affecting athletes’ access to personalized nutrition services and their ability to integrate dietary recommendations into their training regimes [35,73].

Understanding these socio-cultural, historical, and organizational factors is crucial for interpreting the findings of Theme 2 and clarifying the broader context in which athletes engage with personalized nutrition plans. Recognizing the diverse influences on athletes’ perceptions and experiences allows researchers and practitioners to develop more customized and effective approaches to promoting optimal nutrition and performance outcomes within collegiate sports environments.

### 4.3. Theme 3: Challenges and Barriers to Implementation

#### 4.3.1. Theme 3 Interpretation of the Findings

Theme 3, “Challenges and Barriers to Implementation”, identifies the obstacles that hinder athletes’ adherence to personalized nutrition plans in collegiate sports settings. Data analysis from participant interviews revealed several key challenges that athletes face when trying to integrate personalized nutritional interventions into their training and competition routines. The participants reported various barriers, such as time constraints, conflicting dietary advice, financial limitations, and social influences, which impede their ability to engage with and benefit from personalized nutrition plans. These findings highlight the interaction between individual, interpersonal, and environmental factors that influence athletes’ dietary behaviors and decision-making processes within collegiate sports environments.

#### 4.3.2. Theme 3 Comparison with the Existing Literature

The findings from Theme 3, “Challenges and Barriers to Implementation”, highlight the variety of obstacles athletes face when trying to adhere to personalized nutrition plans. These challenges are consistent with the existing literature that identifies time constraints, conflicting dietary advice, financial limitations, and social influences as significant barriers to optimal nutrition among athletes in various sports contexts [74]. This study, however, uniquely focuses on Division III collegiate athletes, whose experiences are shaped by the specific socio-cultural dynamics of their athletic programs and academic institutions.

The study details the specific challenges Division III athletes face, shedding light on the interaction between these barriers and the broader organizational and cultural context of their environments. For instance, time constraints may be more severe due to academic demands or limited access to nutritional resources in smaller athletic programs. Conflicting dietary advice may arise from the varied sources of information athletes consult, such as teammates, coaches, and online platforms [75]. Financial limitations intersect with broader socio-economic factors, affecting athletes’ access to nutritious foods or professional nutritional guidance [34].

Additionally, the study illustrates how these barriers often interact, compounding their effects on athletes’ dietary behaviors. Financial constraints can intensify difficulties in accessing nutritious foods while conflicting dietary advice can lead to confusion and uncertainty about effective nutritional strategies [76]. Social influences, including peer norms or cultural attitudes toward food and body image, further influence athletes’ dietary choices and their adherence to nutrition plans [77].

#### 4.3.3. Theme 3 Contextualization

The findings from Theme 3, which focuses on the challenges and barriers to implementing personalized nutrition plans among Division III collegiate athletes, are shaped by various socio-cultural, historical, and organizational factors inherent to collegiate sports environments. A key contextual factor is the culture within collegiate athletics programs, which may prioritize performance outcomes over holistic athlete well-being or lack the resources to support athletes’ nutritional needs [78]. Additionally, historical trends in sports nutrition and coaching practices might influence athletes’ attitudes toward nutrition and the availability of evidence-based nutritional guidance within collegiate sports settings [57].

Organizational factors within Division III athletic programs, such as the size and structure of the athletic department, can impact athletes’ access to nutritional resources and support services. Smaller programs with limited budgets might struggle to provide comprehensive nutritional education or individualized dietary counseling, leading to disparities in nutritional knowledge and support across different sports teams [34,79]. The socio-economic backgrounds of Division III athletes and their institutions may affect access to nutritious foods, with athletes from lower-income backgrounds facing more significant challenges in maintaining optimal dietary habits [80].

The broader socio-cultural context, including societal norms, cultural attitudes toward food, and prevailing dietary trends, may also affect athletes’ dietary behaviors and adherence to nutrition plans. Societal pressures to achieve certain body standards or conform to restrictive dietary trends may promote disordered eating behaviors or lead to misinformation about nutrition among athletes [81,82]. Cultural attitudes toward food within the athletic community, such as glorifying restrictive eating or normalizing unhealthy dietary habits, may further complicate athletes’ relationships with food and nutrition [83].

Overall, these socio-cultural, historical, and organizational factors contextualize the challenges and barriers identified in Theme 3, influencing the interpretation and implications of the findings. Understanding these factors is crucial for developing targeted interventions and support strategies that address the specific needs and challenges faced by Division III collegiate athletes in adhering to personalized nutrition plans within their sports environment. By considering these factors, researchers and practitioners can devise more effective strategies to promote optimal nutrition and overall well-being among collegiate athletes.

### 4.4. Theme 4: Influence of Team Culture and Environment

#### 4.4.1. Theme 4 Interpretation of the Findings

Theme 4 explores how team culture and environment influence athletes’ attitudes toward nutrition plans, revealing that social dynamics and organizational structures within collegiate sports teams significantly impact athletes’ dietary behaviors and adherence to personalized nutrition plans. This theme emphasizes the role of peer influences, coaching practices, and team norms in shaping athletes’ dietary choices and habits. Athletes often rely on their peers and team leaders for guidance and validation in aspects of athletic performance, including nutrition. The findings indicate that team dynamics, such as peer support for certain dietary practices or the normalization of specific eating behaviors, significantly influence athletes’ dietary choices and adherence to nutrition plans. Nutrition-related behaviors may also become a part of social identity within the team, leading to conformity with team norms or pressure to adhere to specific dietary practices.

Coaching practices and team leadership are critical in shaping attitudes toward nutrition. Coaches who emphasize nutrition education, provide nutritional resources, and integrate nutrition into team culture can positively affect athletes’ awareness and engagement with nutrition plans. Conversely, a lack of support or guidance from coaches may hinder athletes’ adoption of healthy dietary habits. The organizational structure and resources within collegiate sports teams also affect athletes’ access to nutritional support and guidance. Larger programs with dedicated sports nutrition staff or established support systems can offer comprehensive nutrition education, individualized dietary counseling, and access to sports-specific nutritional resources. In contrast, smaller programs with fewer resources may struggle to prioritize nutrition, leading to disparities in nutritional support across different teams. Recognizing these influences allows sports organizations and coaching staff to implement strategies that promote a supportive team culture, prioritizing athletes’ nutritional needs and facilitating adherence to personalized nutrition plans. Targeted interventions and educational initiatives can also empower athletes to make informed dietary choices and optimize their performance within the team environment.

#### 4.4.2. Theme 4 Comparison with the Existing Literature

The findings from Theme 4 align with the existing literature which emphasizes the influence of team culture and environment on athletes’ dietary behaviors and attitudes toward nutrition plans. Previous research has consistently shown the importance of social dynamics within sports teams in shaping athletes’ behaviors and perceptions, particularly regarding nutrition. Studies have identified that peer influences, team norms, and social identity processes are key in determining athletes’ dietary choices and adherence to nutrition plans [84,85].

Additionally, the literature has recognized the role of coaching practices and team leadership in promoting nutrition education and support for athletes. Effective communication and guidance from coaches regarding nutrition are associated with better dietary behaviors and performance outcomes among athletes [86,87]. Conversely, inadequate nutrition support or misinformation from coaching staff has been shown to negatively affect athletes’ nutritional practices and performance [53,88].

The findings also highlight the impact of organizational factors, such as resources and support systems within sports programs, on athletes’ access to nutritional services and support. Research indicates that disparities in access to sports nutrition services exist across different sports organizations, with larger programs typically offering more comprehensive support than smaller programs [79,89]. These disparities can lead to variations in athletes’ nutritional knowledge, behaviors, and performance outcomes depending on the resources available within their teams or programs.

This study addresses some gaps in existing research. While previous studies have highlighted the influence of team culture and coaching practices on athletes’ nutrition-related behaviors [90,91], fewer have examined how these factors impact attitudes toward personalized nutrition plans. This study adds to this understanding by exploring how social dynamics, coaching practices, and organizational factors collectively shape athletes’ perceptions and adherence to personalized nutrition plans within collegiate sports teams.

Additionally, the study contributes by emphasizing the importance of considering socio-cultural, historical, and organizational factors in interpreting athletes’ attitudes toward nutrition plans [92,93]. By placing the findings within the broader socio-cultural and organizational context of collegiate sports, the study provides insights into the complex interactions between individual, interpersonal, and environmental factors influencing athletes’ dietary behaviors and performance outcomes.

The findings from Theme 4 are consistent with the existing literature on the influence of team culture, coaching practices, and organizational factors on athletes’ attitudes toward nutrition plans. However, the study provides new insights into the specific mechanisms through which these factors affect athletes’ perceptions and adherence to personalized nutrition plans within collegiate sports teams, offering a more comprehensive understanding of the role of team environments in promoting athletes’ nutritional health and performance.

#### 4.4.3. Theme 4 Contextualization

The findings of Theme 4, which focus on the influence of team culture and environment on athletes’ attitudes toward nutrition plans, are shaped by various socio-cultural, historical, and organizational factors within collegiate sports settings. The socio-cultural context of collegiate athletics, including norms, values, and beliefs surrounding performance, health, and nutrition, significantly influences athletes’ perceptions of nutrition and dietary practices [2]. Historical trends in sports nutrition education and practices within collegiate sports programs have also shaped the current attitudes and behaviors among athletes.

Organizational factors, such as coaching practices, team dynamics, and institutional resources, critically influence athletes’ nutrition-related experiences [34]. Coaches, as key influencers within the team environment, may impart nutritional knowledge, provide support, and establish norms for dietary practices [94,95]. The availability of sports nutrition services, facilities, and resources within programs affects athletes’ access to nutrition support and guidance [96], and disparities in resources across programs can lead to variations in athletes’ nutritional knowledge, behaviors, and performance outcomes.

Additionally, the broader socio-cultural context of sports and health promotion, including media influences, societal trends, and cultural perceptions of body image and athleticism, shapes athletes’ attitudes toward nutrition plans [97]. Cultural ideals of athletic performance and body aesthetics, as well as media portrayals of nutrition and fitness and trends in diets and wellness, influence athletes’ beliefs and practices related to nutrition [7].

By understanding these socio-cultural, historical, and organizational contexts, we gain a nuanced view of how team culture, coaching practices, institutional resources, and broader influences shape athletes’ attitudes toward nutrition plans and their implementation in collegiate sports environments. This perspective is crucial for developing strategies to promote optimal nutrition practices and support athletes’ performance and well-being within collegiate sports programs.

### 4.5. Theme 5: Suggestions for Improvement

#### 4.5.1. Theme 5 Interpretation of the Findings

Theme 5 explores the participants’ suggestions for improving personalized nutrition plans and includes insights and recommendations provided by the athletes to enhance the effectiveness and accessibility of nutritional interventions in collegiate sports settings. The analysis of the interview data revealed key areas for improvement in nutrition programming, such as personalized guidance, accessibility, education, resources, ongoing support, and accountability, which are all vital for maximizing the impact of nutritional interventions on athletes’ performance and well-being.

Athletes recognized the critical role of nutrition in supporting athletic performance and overall health, expressing a need for more personalized guidance tailored to individual needs, preferences, and goals. They highlighted the variability in nutritional requirements and dietary preferences among athletes and emphasized the need for enhanced nutrition education and resources to help them understand optimal dietary practices and make informed nutritional choices.

Accessibility was identified as a significant concern, with the athletes calling for better access to nutrition services, facilities, and resources within collegiate sports programs. They pointed out challenges related to financial constraints, limited availability of nutrition support services, and inadequate access to nutritious food options, especially for athletes with specific dietary needs.

Additionally, the participants stressed the importance of ongoing support and accountability in maintaining adherence to personalized nutrition plans. They advocated for consistent monitoring, feedback, and follow-up to track progress, address challenges, and adjust nutrition programming as needed. The athletes also noted the role of the team environment and social support networks in promoting positive dietary behaviors and adherence to nutrition plans.

Overall, the findings from Theme 5 underscore the value of athletes’ insights and recommendations for enhancing personalized nutrition plans in collegiate sports settings. By incorporating athletes’ suggestions for improvement, sports programs can better support athletes in optimizing their dietary practices, improving their performance, and promoting their overall health and well-being.

#### 4.5.2. Theme 5 Comparison with the Existing Literature

The findings from Theme 5, which focuses on athletes’ suggestions for improving personalized nutrition plans, align with and contribute to the existing literature on sports nutrition. Previous research has emphasized the importance of personalized nutritional interventions tailored to athletes’ individual needs, preferences, and goals to optimize performance and health [1]. In line with this, the participants in the current study highlighted the need for personalized guidance, education, and resources to support their dietary practices.

The emphasis on accessibility corresponds with previous studies that have identified barriers athletes face in accessing nutrition services and resources within collegiate sports programs [80]. Financial constraints, limited availability of nutrition support services, and inadequate access to nutritious food options are significant challenges for athletes, especially those from disadvantaged backgrounds or with specific dietary requirements [74]. The findings also highlight the importance of ongoing support and accountability in maintaining adherence to nutrition plans consistent with behavior change theory and sports psychology. Research has shown that continuous monitoring, feedback, and follow-up are effective in promoting adherence to dietary recommendations and optimizing performance outcomes among athletes [98].

There may be inconsistencies or gaps between the findings of this study and the existing literature. While the athletes in this study emphasized the role of the team environment and social support networks in promoting positive dietary behaviors, some studies have indicated that team culture and peer influences might also lead to unhealthy eating habits or disordered eating behaviors among athletes [99,100].

Overall, the findings from Theme 5 offer insights into how personalized nutrition plans can be enhanced to better support athletes’ performance and well-being. By incorporating athletes’ suggestions into nutrition programming, sports programs can improve the effectiveness and accessibility of nutritional interventions, ultimately benefiting the health and performance of athletes.

#### 4.5.3. Theme 5 Contextualization

The findings from Theme 5, which focuses on athletes’ suggestions for improving personalized nutrition plans, are influenced by sociocultural, historical, and organizational factors that shape athletes’ dietary behaviors and perceptions of nutrition within collegiate sports environments. Socio-cultural factors like cultural norms, societal expectations, and peer influences significantly shape athletes’ attitudes toward nutrition. In collegiate sports, where teamwork is emphasized, athletes may adopt the dietary habits of their teammates, coaches, and support staff [101,102].

Historically, sports nutrition has shifted from basic dietary recommendations to a more personalized, evidence-based approach that caters to specific athletes’ needs and goals [103]. This change is due to advancements in sports science and nutrition research, recognizing the role of nutrition in enhancing athletic performance and health. Organizational factors such as nutrition support services, resources, and policies within collegiate sports programs also play a role in how athletes engage with and access nutrition programming [103].

The athletes’ suggestions for improvement are reflections of not only their individual preferences and needs but also the broader socio-cultural and organizational contexts they operate within. For instance, calls for more personalized guidance and ongoing support may arise from athletes’ desire for more autonomy in their dietary choices and the need for nutritional strategies that meet their performance goals.

Recommendations for better accessibility and education might be influenced by socio-economic disparities among athletes, differences in nutritional knowledge, and resource availability within collegiate sports programs. By considering these contextual factors, sports programs can better understand and address the specific needs and challenges athletes face regarding nutrition.

Overall, athletes’ perceptions, attitudes, and behaviors toward nutrition are shaped by socio-cultural norms, historical trends in sports nutrition, and organizational structures within collegiate sports programs. Recognizing and addressing these factors can help sports programs develop more effective and culturally sensitive nutritional interventions that enhance athletes’ performance, health, and well-being.

### 4.6. Interpreting Athletes’ Nutrition Behaviors through Ecological Systems Theory (EST)

Situated within the framework of EST, this study provides insights into the complex interactions between individual athletes and their broader socio-cultural and environmental contexts. EST highlights the dynamic interactions between individuals and their environments, underscoring the reciprocal influences that shape human development and behavior [16,17]. In this research, the dietary behaviors and personalized nutrition plans of collegiate athletes are viewed as outcomes of interactions across various ecological systems, including the microsystem (individual), mesosystem (interactions between individuals and immediate environments), exosystem (external influences), and macrosystem (broader cultural and societal influences).

By applying EST, this study enhances the theoretical understanding of athletes’ dietary behaviors by detailing how these behaviors are influenced by a mix of personal factors and social, cultural, and organizational elements within their sports environments. It emphasizes the need to consider the connections between these systems to fully understand athletes’ dietary behaviors. Additionally, the findings bring new perspectives to existing theories by stressing the impact of team culture and environmental factors on athletes’ attitudes and behaviors toward nutrition. While prior research has often focused on individual determinants of dietary behaviors [104], this study highlights the role of contextual factors like team dynamics, coaching practices, and cultural norms in shaping athletes’ dietary choices and their adherence to nutrition plans. Thus, the study extends the application of EST by emphasizing the significance of mesosystem and exosystem influences in understanding the dietary behaviors of athletes in collegiate sports settings.

### 4.7. Reflexivity

As researchers, our perspectives, biases, and assumptions shape the research process and findings [45]. Our backgrounds in sports science and nutrition likely influenced our focus on personalized nutrition plans and their effects on athletes’ performance and well-being. These elements guided our selection of research questions, theoretical frameworks, and interpretation of the findings. Additionally, being directly involved in data collection and analysis, our preconceived notions about the benefits of personalized nutrition might have influenced how we approached and interpreted the data, as well as our interactions with the participants during interviews. Reflexivity, or the process of critically examining our roles and biases in the research process, is crucial for enhancing the credibility and trustworthiness of our findings [45]. We engaged in reflexivity by continuously questioning our assumptions and considering how our backgrounds might affect the study, along with exploring alternative interpretations of the data. This practice allowed us to maintain a critical stance toward the data and be transparent about our roles as researchers, thereby increasing the transparency and rigor of the study. Ultimately, by acknowledging and reflecting on our perspectives and biases, we enhance the credibility and trustworthiness of our research, allowing readers to critically evaluate the findings and interpretations.

### 4.8. Trustworthiness

The validity and trustworthiness of the findings in this research are crucial, and several measures were taken to ensure the study’s rigor and credibility. Data triangulation was employed, involving multiple sources such as semi-structured interviews, observational data, and document analysis to corroborate and cross-validate the findings, thereby reducing bias and enhancing credibility [44]. Member checking was also conducted, whereby the participants reviewed and provided feedback on the emerging themes and interpretations after data analysis, helping to confirm the accuracy of their contributions and address any misunderstandings [42,43].

Peer debriefing provided external perspectives on the research process, identifying potential biases, errors, or oversights by holding regular discussions with colleagues and peers not involved in the study [45]. Additionally, reflexivity was practiced throughout to mitigate the influence of researchers’ biases and assumptions on the findings [46]. By critically examining our roles and perspectives, we ensured that our interpretations were transparent and reflective of the data. These steps, including data triangulation, member checking, peer debriefing, and reflexivity, collectively enhance the study’s credibility and support the reliability and validity of the findings.

### 4.9. Implications

The implications of this study for theory, practice, policy, and future research are substantial. By using Ecological Systems Theory (EST) to analyze athletes’ nutrition behaviors, the study sheds light on how individual, interpersonal, and environmental factors influence dietary practices. These insights advance the field of sports nutrition by underscoring the importance of considering socio-cultural, organizational, and environmental contexts in creating effective nutritional interventions for athletes. Practically, these findings help in developing tailored nutrition plans that address athletes’ specific needs and challenges, aiding practitioners in providing personalized nutrition guidance to optimize performance, health, and well-being. They also highlight the role of a supportive team culture in promoting healthy dietary practices. On a policy level, the study points to the necessity for comprehensive nutrition education programs and policies within sports organizations to foster evidence-based practices and overcome adherence barriers. Additionally, it advocates for integrating nutrition into broader athlete development initiatives to support holistic well-being. These insights are valuable not only in sports nutrition but also for practitioners, policymakers, and researchers focused on enhancing athlete health and performance.

### 4.10. Study Limitations

While this study provides insights into athletes’ nutrition behaviors, it is important to recognize its limitations. The use of qualitative interviews primarily limits the generalizability of the findings beyond the specific collegiate sports context studied. Recruiting participants from a single environment may restrict the applicability of the results to athletes in other settings or at different competitive levels. Additionally, the subjective nature of qualitative research introduces the potential for researcher bias and discrepancies in interpretation despite efforts to maintain reflexivity and rigor.

The reliance on self-reported data may also affect the reliability and accuracy of the responses due to recall or social desirability biases. The study’s qualitative design further limits the ability to establish causal relationships between variables. Furthermore, the exclusion of perspectives from coaches, nutritionists, and other key stakeholders creates a gap, reducing the depth of understanding of the socio-cultural and organizational influences on athletes’ nutrition behaviors. Despite these challenges, acknowledging and discussing these limitations transparently helps refine the interpretation of the findings and guides future research to address these issues.

### 4.11. Future Research

Future research could examine the long-term effects of personalized nutrition plans on athletes’ performance, health outcomes, and adherence behaviors. Longitudinal studies that track athletes over time could reveal the sustainability and effectiveness of these interventions. Additionally, exploring the role of technology, such as mobile apps or wearable devices, could provide practical solutions for overcoming barriers like time constraints and conflicting dietary advice as identified in this study. Research into how socio-cultural and organizational factors such as sex, ethnicity, and team dynamics affect athletes’ dietary behaviors could offer a more comprehensive view of the influences at play. Furthermore, incorporating interdisciplinary approaches with input from nutritionists, psychologists, and sports scientists could deepen our understanding of athletes’ complex nutritional needs and behaviors. Exploring the perspectives of coaches, athletic trainers, and other stakeholders in collegiate sports could also provide insights into the organizational context that influences athletes’ dietary behaviors and the implementation of personalized nutrition plans.

## 5. Conclusions

In conclusion, this study has explored the experiences and perceptions of Division III athletes regarding personalized nutrition plans. Through qualitative interviews, five principal themes emerged, illustrating the nuanced landscape of collegiate athletic nutrition. These themes include (1) Nutritional Knowledge and Awareness, where participants showed varied understanding of personalized nutrition plans; (2) Perceived Benefits of Personalized Nutrition Plans, with the athletes noting improved performance and well-being; (3) Challenges and Barriers to Implementation, highlighting obstacles such as time constraints and conflicting dietary advice; (4) Influence of Team Culture and Environment, demonstrating how team dynamics and cultural norms shape nutritional approaches; and (5) Suggestions for Improvement, where the athletes provided actionable insights to enhance the effectiveness of nutrition plans.

The findings underscore the need for sports nutritionists and coaches to design nutritional interventions sensitive to the diverse backgrounds and individual circumstances of Division III athletes. Additionally, the study advocates for enhanced nutritional education and support systems that can empower athletes to overcome barriers to effective nutrition plan adherence.

By contributing to both the theoretical frameworks and practical strategies in sports nutrition, this research paves the way for future studies to examine specific interventions to improve personalized nutrition plans’ adoption and impact. This will lead to refined, evidence-based practices that enhance the health and performance outcomes of Division III athletes, ensuring that nutritional strategies are both effective and inclusive.

## Figures and Tables

**Figure 1 healthcare-12-00923-f001:**
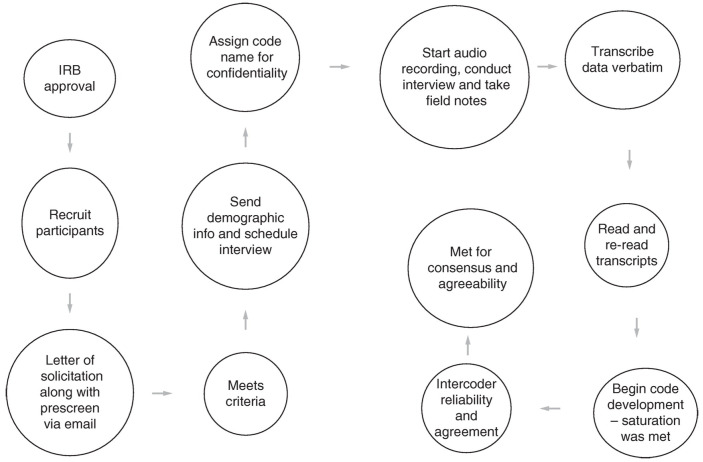
Interview process.

**Figure 2 healthcare-12-00923-f002:**
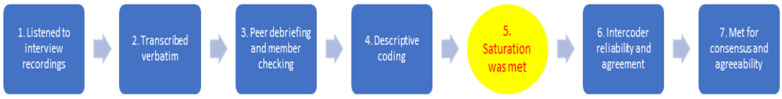
Data analysis process.

**Table 1 healthcare-12-00923-t001:** Interview questions.

Question	Probes
1. Describe your current dietary practices as a Division III athlete.	- What types of foods do you typically consume daily?
	- How do you plan your meals around your athletic schedule?
2. Explain when you participated in a personalized nutrition plan tailored specifically for your athletic performance, if any at all.	- If yes, could you describe your experience with the personalized plan?
	- What prompted you to participate in a personalized nutrition plan?
3. What are some perceived benefits of personalized nutrition plans for athletes like yourself?	- Have you noticed any improvements in your athletic performance as a result of the personalized nutrition plan?
	- How do you think personalized nutrition plans contribute to your overall well-being?
4. Conversely, what challenges or limitations have you encountered with personalized nutrition plans?	- Have you faced any difficulties adhering to the personalized nutrition plan?
	- How do external factors, such as budget or access to specific foods, impact your ability to follow the plan?
5. How do you think personalized nutrition plans compare to more general dietary guidelines provided for athletes?	- In what ways do personalized nutrition plans differ from generic dietary guidelines?
	- Do you believe personalized plans are more effective in optimizing athletic performance?
6. Share any specific examples or anecdotes illustrating the impact of personalized nutrition plans on your athletic performance or overall well-being, if any at all.	- Can you recall when following the personalized nutrition plan positively affected your athletic performance?
	- How do you measure the success or effectiveness of the personalized plan?
7. How does nutrition affect your overall athletic performance and recovery?	- How do you prioritize nutrition within your training regimen?
	- How do you adjust your nutrition plan during intense training or competition?
8. How do you perceive the importance of individualized nutrition guidance compared to other aspects of your athletic training?	- Do you believe personalized nutrition plans are as important as other aspects, such as strength training or conditioning?
	- How do you think personalized nutrition contributes to your overall athletic development?
9. Explain when you sought advice or guidance from nutrition professionals or experts regarding your dietary habits as an athlete, if any at all.	- If yes, what was your experience like interacting with nutrition professionals?
	- How did their recommendations align with your personal preferences and goals?
10. What recommendations would you offer for improving personalized nutrition programs for Division III athletes?	- Do you want to enhance or modify any specific aspects of the personalized nutrition plan?
	- How do you think personalized nutrition programs could better meet the needs of Division III athletes?
11. How do you believe your experiences with personalized nutrition plans may differ from those of athletes in higher divisions or professional leagues?	- Do you perceive any unique challenges or opportunities for Division III athletes in implementing personalized nutrition plans?
	- How do you think personalized nutrition plans may vary across different levels of competition?
12. What are the key takeaways or lessons from your experiences with personalized nutrition plans as a Division III athlete?	- How have your nutrition and athletic performance perspectives evolved by participating in personalized nutrition programs?
	- What advice would you offer other Division III athletes considering adopting personalized nutrition plans for improved performance?

**Table 2 healthcare-12-00923-t002:** Themes.

Theme	Description	Number of Participants (n)	Number of Transcript Excerpts (n)
1. Nutritional Knowledge and Awareness	Participants demonstrated varying levels of understanding regarding personalized nutrition plans.	27	105
2. Perceived Benefits of Personalized Nutrition Plans	Athletes expressed positive outcomes such as improved performance and overall well-being.	29	115
3. Challenges and Barriers to Implementation	Various obstacles, including time constraints and conflicting dietary advice, hindered plan adherence.	22	80
4. Influence of Team Culture and Environment	Team dynamics and cultural norms significantly influenced athletes’ attitudes toward nutrition plans.	24	90
5. Suggestions for Improvement	Participants provided valuable insights and recommendations for enhancing the effectiveness of plans.	20	70

**Table 3 healthcare-12-00923-t003:** Theme 1 Codes: Nutritional Knowledge and Awareness.

Code	Number of Participants (n)	Number of Transcript Excerpts (n)
Understanding	25	90
Awareness	20	80
Nutritional Knowledge	22	85
Misconceptions	15	60
Confusion	10	40
Interest	18	70

**Table 4 healthcare-12-00923-t004:** Theme 2 Codes: Perceived Benefits of Personalized Nutrition Plans.

Code	Number of Participants (n)	Number of Transcript Excerpts (n)
Improved Athletic Performance	23	90
Enhanced Recovery	19	75
Increased Energy Levels	15	60
Better Overall Well-being	18	70
Optimal Body Composition	12	50
Mental Clarity and Focus	14	55
Reduced Risk of Injury	11	45
Enhanced Immune Function	9	35
Improved Sleep Quality	10	40
Greater Consistency in Performance	13	55

**Table 5 healthcare-12-00923-t005:** Theme 3 Codes: Challenges and Barriers to Implementation.

Code	Number of Participants (n)	Number of Transcript Excerpts (n)
Time Constraints	18	65
Conflicting Advice	16	55
Accessibility	12	42
Financial Concerns	10	35
Lack of Support	8	28
Motivation	7	24
Dietary Preferences	6	21
Information Overload	5	18
Social Influence	4	14
Environmental Factors	3	11

**Table 6 healthcare-12-00923-t006:** Theme 4 Codes: Influence of Team Culture and Environment.

Code	Number of Participants (n)	Number of Transcript Excerpts (n)
Team Dynamics	21	75
Coach Influence	18	65
Peer Influence	20	70
Cultural Norms	15	55
Social Support	17	60

**Table 7 healthcare-12-00923-t007:** Theme 5 Codes: Suggestions for Improvement.

Code	Number of Participants (n)	Number of Transcript Excerpts (n)
Access to Nutritional Resources	15	55
Education and Awareness	18	60
Individualized Support	12	45
Team-Based Approach	10	40
Time Management	14	50

## Data Availability

Data are available upon request by contacting the corresponding author, James Stavitz, jstavitz@kean.edu.

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
