# Peer review of "Exploring the Experiences and Perspectives of Division III Athletes Regarding Personalized Nutrition Plans for Improved Performance—A Qualitative Investigation"

_healthcare, 2024, doi:10.3390/healthcare12090923_

Round 1

Reviewer 1 Report

Comments and Suggestions for Authors

I have carefully reviewed the article and would like to provide constructive feedback to enhance the quality and clarity of the work. Please find below a summary of my observations and suggestions for improvement:

Abstract:

The abstract is well-written and clear, No additional comments.

Introduction:

Lines 50-51: what are the differences between elite athletes and those from the III division that justify specific nutrition studies related to this population?

Lines 58-71: this paragraph lacks references to support the assertion that athletes from the III division require differentiated investigation. They need to perform at their best, with similar schedules and training to elite athletes. Do III division athletes have greater demands than elite athletes?

The topic "Ecological Systems Theory" seems disconnected from the introduction. It is unclear why this theory is being discussed at this point in the text.

Methods:

Lines 135-137: Do all sports present similar behavioral characteristics in this population? Were all sports from the III division included?

Lines 151-155: was this a selection criterion? How can we ensure that the sample was not subjectively and selectively chosen, resulting in a serious bias to the study?

How was the sample size calculated and defined? Was the final number of participants significant?

Results:

Which sports are referred to in lines 300-301?

Line 319: Could you please clarify if these five themes were defined after data collection?

Discussion:

Attention to the spelling error throughout the discussion: "Comarpsion with Exisitng Literautre"

Please clarify the necessity of the topic "Reflexivity." In my opinion, this topic exceeds the study's objectives without providing information derived from the results obtained.

Conclusion:

The conclusion does not align with the study's objectives. The way it is presented summarizes the study, including considerations and reflections from the discussion, which is inappropriate. This section needs to be rewritten clearly and related only to the proposed objectives and results of the study.

Reviewer 2 Report

Comments and Suggestions for Authors

This study delves into the perceptions and experiences of collegiate athletes regarding personalized nutrition plans within collegiate sports settings. The study utilized semi-structured interviews to gather rich and in-depth data. Thematic analysis was employed to dissect interview transcripts, enabling the identification of recurring themes and patterns.

The abstract is too long it should have a maximum of 200 words, in addition, the division of sections is inconsistent with the requirements of the journal.

In-text citations should be given in parentheses.

The introduction should include the stated objectives of the study at the end, I think this part of the study should be reworked to be more structured. Describe the EST beforehand, and indicate the objective and hypothesis at the end.

112-123 I believe that such a description should be addressed in the discussion of the study. In this section, a specific study design should be given and not a broad justification for its choice.

The entire study is 39 pages, I think the study is interesting however the structure is incorrect. The methodology of the study by its volume and form of description is unspecific.

Table 8 should be considered for inclusion as supplementary material.

The conclusions should contain the most important findings of the study and not a description of what was done in the discussion section.

I believe that the study in its current form does not meet the criteria of a journal. The volume should be shortened, the information should be systematized, and it should be limited to the essential elements.

Round 2

Reviewer 2 Report

Comments and Suggestions for Authors

Thank you for following my suggestions. I think the work is suitable for publication.